# Adenovirus-Neutralizing and Infection-Promoting Activities Measured in Serum of Human Brain Cancer Patients Treated with Oncolytic Adenovirus Ad5-∆24.RGD

**DOI:** 10.3390/ijms26020854

**Published:** 2025-01-20

**Authors:** Ida H. van der Meulen-Muileman, Joana Amado-Azevedo, Martine L. M. Lamfers, Anne Kleijn, Sander Idema, David P. Noske, Clemens M. F. Dirven, Victor W. van Beusechem

**Affiliations:** 1Amsterdam UMC location Vrije Universiteit Amsterdam, Medical Oncology, De Boelelaan 1117, 1081 HV Amsterdam, The Netherlands; 2Department of Neurosurgery, Brain Tumor Center, Erasmus University Medical Center, Dr. Molewaterplein 40, 3015 GD Rotterdam, The Netherlands; 3Amsterdam UMC location Vrije Universiteit Amsterdam, Neurosurgery, De Boelelaan 1117, 1081 HV Amsterdam, The Netherlands; 4Brain Tumor Center Amsterdam, 1081 HV Amsterdam, The Netherlands; 5Cancer Center Amsterdam, Cancer Biology and Immunology, 1081 HV Amsterdam, The Netherlands; 6Amsterdam Infection and Immunity Institute, Cancer Immunology, 1081 HV Amsterdam, The Netherlands; 7ORCA Therapeutics BV, Onderwijsboulevard 225, 5223 DE ‘s-Hertogenbosch, The Netherlands

**Keywords:** oncolytic adenovirus, virotherapy, clinical trial, glioblastoma, virus neutralization, modified fiber knob, RGD motif

## Abstract

Oncolytic adenoviruses derived from human serotype 5 (Ad5) are being developed to treat cancer. Treatment efficacy could be affected by pre-existing or induced neutralizing antibodies (NAbs), in particular in repeat administration strategies. Several oncolytic adenoviruses that are currently in clinical development have modified fiber proteins to increase their infectivity. One example is Ad5-∆24.RGD, which carries a cyclic RGD peptide insert in the fiber protein to allow cell entry via integrins. The effect of anti-Ad5 NAbs on anticancer efficacy could be different for oncolytic adenoviruses with RGD-modified fibers than for unmodified Ad5-based viruses. Here, we determine pre-existing and elicited NAb titers in the serum of patients with glioblastoma who were treated by delivering Ad5-∆24.RGD to the tumor and to the surrounding tumor-infiltrated brain. We show that intracranial infusion of Ad5-∆24.RGD induced mainly neutralization of adenovirus native tropism. Infection of cells with RGD-modified virus was significantly less affected. In cerebrospinal fluid, neutralizing activity against RGD-mediated infection remained very low. Thus, the RGD-mediated alternative cell entry route allowed to bypass pre-existing and induced anti-Ad5 neutralization. Interestingly, in the course of these experiments, we discovered that the serum of most humans promotes the uptake of RGD-modified adenovirus in human cells. The until now unidentified infection-stimulating factor seems distinct from serum proteins known to promote Ad5 infection. Together, our work supports the utility of RGD-modified oncolytic adenoviruses for the treatment of cancer in humans. Since these viruses hardly induced neutralization, they seem particularly suitable for repeat administration treatments.

## 1. Introduction

Oncolytic adenoviruses have shown substantial preclinical and clinical efficacy as anticancer agents [1,2]. These viruses selectively replicate in and kill cancer cells, thereby eliciting antitumor immune responses. Commonly used oncolytic adenoviruses are derived from human adenovirus serotype 5 (Ad5). In addition to natural polyreactive IgM antibodies that bind to Ad5 with low affinity, promoting clearance from the circulation [3,4], most adult humans have high-affinity anti-Ad5 neutralizing IgG antibodies (NAbs) as a consequence of natural exposure to Ad5 earlier in life [5]. This raises the concern that pre-existing NAbs might attenuate the efficacy of oncolytic adenovirus treatment. In addition, humoral immune responses elicited against administered oncolytic adenovirus are anticipated to increase neutralization activity, thus hampering effective repeat administration.

Several studies have addressed the influence of NAbs on adenovirus treatment efficacy in rodent models. The results of these studies did not decisively support or refute the concerns. While one study showed that the efficacy of intravenous oncolytic adenovirus treatment was not affected by infusion of anti-Ad5 NAb-containing human serum [6], another study reported that high anti-Ad5 NAb titers inhibited tumor growth inhibition by intratumoral injected oncolytic adenovirus [7].

In order to circumvent anti-Ad5 pre-existing immunity in oncolytic virotherapy or vaccination, several strategies were proposed. These include the use of different human adenovirus serotype vectors [5,8], chimeric Ad5-derived viruses packaged in capsids with proteins from different serotype adenoviruses [9,10], or adenoviruses from different species origins [11,12,13]; and shielding Ad5 capsids from recognition by NAbs by providing them with an engineered protective coat [14,15,16,17].

All three major adenovirus capsid proteins, hexon, penton base and fiber, have been proposed to contain important immune neutralization epitopes [18,19,20]. Several oncolytic adenoviruses that are currently in clinical development have modified capsids to increase their infectivity. This includes, amongst others, Ad5-based viruses with a chimeric Ad5/3 fiber protein to redirect entry via Ad3 receptors [21], a cyclic RGD peptide insert in the fiber protein to allow cell entry via integrins [22,23,24,25] or with a polylysine (pK7) insert in the fiber protein targeting heparan sulfate proteoglycans [26]. Conceivably, the efficacy of these fiber-modified viruses might be less hampered by anti-Ad5 fiber NAbs. However, the effects of NAbs binding to other capsid proteins are probably not altered by these modifications.

The relative importance of fiber-dependent and fiber-independent neutralization processes is largely unknown. Furthermore, treatment with fiber-modified viruses could elicit differential inhibition of infection via native and alternative virus receptor recognitions. Therefore, the consequences of pre-existing or induced anti-Ad5 NAbs for the efficacy of single or repeat administration treatments with infectivity-enhanced oncolytic adenoviruses could be different from what is predicted on the basis of studies using unmodified Ad5-based viruses. This was addressed in studies with immunized mice [27,28] and human cancer patients [29]. Generally, wild-type Ad5 was more immunogenic than capsid-modified viruses. NAb titers were highest against the injected virus, but also NAb titers against viruses with other capsid variants rose, suggesting cross-neutralization.

Here, we contribute to these investigations by analyzing NAb titers in the serum of patients with recurrent glioblastoma who were infused with oncolytic adenovirus Ad5-∆24.RGD via convection-enhanced delivery (CED) to the tumor and to the surrounding tumor-infiltrated brain in a completed clinical trial (NCT01582516) [24]. Pre-existing and elicited neutralization was determined using a native Ad5 vector and an RGD-modified Ad5 vector to distinguish between neutralization of the two alternative cell entry pathways used by Ad5-∆24.RGD. We show that intra- and peritumoral infusion of Ad5-∆24.RGD induced mainly neutralization of adenovirus native tropism. In cerebrospinal fluid (CSF), neutralizing activity against RGD-mediated infection remained very low. Interestingly, in the course of these experiments, we discovered that the blood serum of most humans promotes, in particular, RGD-mediated uptake of recombinant adenovirus in human cells. The infection-stimulating serum factor seems distinct from serum proteins known to promote Ad5 infection.

## 2. Results

### 2.1. Neutralizing Antibody Titers in Patients with Glioblastoma Undergoing Treatment with Oncolytic Adenovirus Ad5-∆24.RGD

Nineteen patients with recurrent glioblastoma were treated in a phase I dose-escalation study using oncolytic adenovirus Ad5-∆24.RGD [24]. Virus doses of 10^7^, 10^8^, 10^9^, 10^10^ or 3 × 10^10^ genome copies (gc) were administered to the tumor and surrounding brain by CED. Serum samples were collected before virus infusion and at 2 and 4 weeks after treatment. Previously, total anti-Ad5 IgM and IgG antibodies were determined in the pre-infusion and week-4 samples using ELISA [24]. To investigate if the antibodies were capable of neutralizing adenovirus particles and could thus hamper therapeutic efficacy, we determined pre-existing and elicited NAb titers on serum samples from this clinical trial by measuring inhibition of adenovirus vector-mediated Firefly luciferase gene transfer to A549 cells. Because Ad5-∆24.RGD infects cells via two alternative entry pathways, i.e., via its native tropism for the high-affinity coxsackievirus and adenovirus receptor and via an interaction of the cyclic RGD motif inserted in its fiber capsid protein with cell surface integrins [30], inhibition of these alternative cell entry pathways was investigated separately. To this end, two different adenovirus vectors were used, i.e., AdLuc [31] with native tropism and AdLucRGD [32] that can infect cells via native as well as cyclic RGD-mediated receptor interactions.

Appendix A show the dilution titrations testing the sera of all 19 patients at the three time-points for neutralization of AdLuc and AdLucRGD, respectively. Figure 1 shows the NAb titers derived from these experiments. As can be seen in Figure 1A, 6 patients did not have detectable pre-existing neutralizing activity against AdLuc or AdLucRGD, and 12 patients had pre-existing NAbs against both viruses. While one patient had pre-existing neutralizing activity only against AdLucRGD, neutralization titers against this virus were generally, although not significantly, lower (*p* = 0.055). Upon infusion of Ad5-∆24.RGD in and around the tumor, serum NAb titers rose prominently, in particular between weeks 2 and 4 (Figure 1B,C). This rise was only significant for the neutralization of AdLuc (*p* = 0.009; AdLucRGD: *p* = 0.096). Consequently, 4 weeks after Ad5-∆24.RGD infusion, median NAb titers against AdLuc were 16-fold higher than against AdLucRGD (Figure 1D; *p* = 0.003). NAb titers did not correlate with previously determined total anti-Ad5 IgG titers (Appendix A). Apparently, total IgG titers are poor predictors of neutralization activity in these patients. The increases in NAb titers were not Ad5-∆24.RGD dose-dependent (Appendix A). Hence, immunization by intracranial infusion of Ad5-∆24.RGD at the used doses reproducibly induced mainly neutralization of adenovirus native tropism.

Previously, we measured total anti-adenovirus IgG titers in the serum and cerebrospinal fluid of these patients. While antibody titers are usually low in healthy brains, we found that the CSF in these patients contained anti-adenovirus IgG antibodies at considerable titers not much lower than were measured in serum [24]. Therefore, we also investigated neutralizing activities in cerebrospinal fluid. Matching serum and CSF samples from selected patients with high NAb titers in their peripheral blood after 4 weeks were analyzed for neutralization of AdLuc and AdLucRGD. As can be seen in Figure 2, AdLuc NAb titers were significantly lower in CSF than in blood (median 64-fold reduced; *p* = 0.034). AdLucRGD NAb titers were also considerably lower in CSF than in blood, but because these titers were already low in blood, the difference did not reach significance (16-fold; *p* = 0.055). The greatly reduced NAb titers in CSF compared with blood contrasted with the similar total anti-adenovirus IgG titers that were measured in these fluids before [24].

### 2.2. Detection of a Blood Component That Promotes RGD-Mediated Adenovirus Infection

In the course of the NAb titration experiments, we observed that many serum samples enhanced adenovirus infection compared with serum-free controls. Whilst most samples exhibited a typical NAb dose response, with inhibition of infection at high serum concentration that is lost upon dilution, several samples exhibited a different pattern. At certain serum dilutions, infection efficiencies exceeded that of serum-free controls (Appendix A). Figure 3A shows exemplary observations made with AdLucRGD on sera from patient #8. These sera exhibited peak infection efficiencies well above 100%. Pre-infusion, the serum of patient #8 did not contain detectable NAbs. Consequently, high AdLucRGD infection efficiencies were seen at high serum concentrations. After Ad5-∆24.RGD infusion, however, emerged NAbs inhibited AdLucRGD infection at high serum concentration. Upon depleting the samples of NAbs by dilution, the infection-promoting effect became detectable. In all cases, the infection-promoting activity was lost upon further serum dilution, with infection efficiencies dropping to approximately 100%, i.e., comparable to the serum-free control, at low serum concentrations. This pattern suggested that the serum contained a molecule with concentration-dependent infection-promoting activity.

To identify human serum samples that promoted adenovirus infection, taking the variation in the data into consideration, we set an arbitrary threshold at 200% infection efficiency compared with controls without human serum. Appendix A shows the serum dilution titration results of all 34 samples that exceeded this threshold at any dilution. Figure 3B shows the peak infection efficiencies measured for all serum samples. As can be seen, the serum of 5/19 (26%) patients promoted AdLuc infection, and the serum of 15/19 (79%) patients promoted AdLucRGD infection at least at one of the sampling points. Infection enhancement was stronger for AdLucRGD than for AdLuc (*p* < 0.0001; paired *t*-test) and was seen before as well as after Ad5-∆24.RGD infusion (in 10 and 14 patients, respectively). Interestingly, only 3 of the 57 tested serum samples promoted infection by AdLuc only, whereas many serum samples (27/57) promoted infection by AdLucRGD but not by AdLuc. Thus, these sera most probably contained a factor that promoted, in particular, RGD-mediated adenovirus infection.

To investigate if the factor was associated with glioblastoma, we analyzed human plasma from healthy donors for the promotion of infection with AdLucRGD. As can be seen in Figure 3C, 8 of 14 samples (57%) increased infection at least 2-fold. Thus, the infection-stimulating factor was present in the blood of a similar proportion of healthy donors as was observed for patients with glioblastoma before adenovirus administration (10/19; 53%). Thus, the factor probably is a natural component of human blood.

### 2.3. Partial Characterization of the Blood Component That Promotes RGD-Mediated Adenovirus Infection

All NAb titration experiments were performed with heat-inactivated serum or CSF to inactivate the complement system. This was done to avoid antibody-dependent complement activation [33] and opsonization of adenovirus particles with complement factors, which neutralizes the virus [34,35] and could thus confound NAb quantification. Apart from the intended complement inactivation, heating can also change serum protein composition [36], promote the formation of immunoglobulin aggregates [37] and influence the cellular uptake of nanoparticles [36,38]. Therefore, we compared AdLucRGD infection in the presence of serum samples from eight Ad5-∆24.RGD-treated glioblastoma patients that were either left untreated or were heat-inactivated. In addition, since blood coagulation factors are known to influence Ad5 infection [39], we also compared the presence of the infection-promoting factor in plasma versus serum. This was done by including matching plasma samples that were available from four of the eight selected patients. As can be seen in Figure 4, mean infection efficiencies were above 200% compared with medium controls for all four sample types. Notably, infection of A549 cells with AdLucRGD was stimulated similarly by the selected serum and plasma samples but was higher when serum was heat-inactivated (*p* = 0.034 compared with untreated serum and *p* = 0.069 compared with heat-inactivated plasma).

To separate infection-promoting and neutralizing activities, we fractionated serum samples by size exclusion. For this, we used two samples from patient #7 collected on days 3 and 4 after oncolytic adenovirus infusion that contain NAbs as well as the putative RGD-mediated infection-promoting factor. Individual elution fractions were scanned for their effect on the transduction of A549 cells with AdLucRGD at low sample dilution (Figure 5A,B). Neutralizing activity and infection-promoting activity were observed in different fractions. Early and late eluting fractions (containing molecules with high and low MW, respectively) revealed infection-promoting activity; neutralization was seen in fractions representing mid-weight molecules. Next, dilution titrations were performed with selected fractions containing infection-promoting and neutralizing activities in comparison to the complete input serum used for fractionation (Figure 5C,D). As can be seen, the medium MW fractions of both samples neutralized AdLucRGD at approximately 3–5-fold reduced efficacy compared with input samples. In contrast, the selected low and high-MW fractions stimulated infection. In these fractions, infection-promotion was 2–3-fold stronger than observed for unfractionated serum, suggesting that these fractions were depleted of NAbs. Size exclusion thus separated the infection-stimulating and neutralizing activities. Finally, to test our assumption that the medium MW fractions that inhibited infection with AdLucRGD contained NAbs, we depleted these fractions of IgG antibodies using protein-G-coated beads (Figure 5E). IgG depletion reduced neutralization, confirming that this was caused by NAbs. Remarkably, IgG depletion of the day-4 sample revealed modest infection-promoting activity also in the medium MW fraction. Hence, infection-promoting activity was found in fractions representing a broad range of molecular weights. Therefore, on the basis of these experiments, the molecular weight of the infection-promoting factor(s) could not be estimated.

## 3. Discussion

Here, we investigated pre-existing and oncolytic adenovirus-induced neutralizing activity in the serum of human patients with glioblastoma. Infection of human cells with RGD-modified virus was hampered less by pre-existing NAbs in the serum of most patients than infection via native Ad5 tropism, although the difference did not reach significance. Administration of an RGD-modified oncolytic adenovirus to tumor and tumor-infiltrated brain induced primarily neutralization of native Ad5 tropism. Neutralization of AdLucRGD capable of infecting cells via both routes was not significantly induced, suggesting that the RGD-mediated component of virus uptake into cells was inhibited less than the natural Ad5 infection process. Our observations are in line with those made in immunization studies in mice, where injection of an RGD-modified Ad5 vector inhibited mainly native Ad5 vector infection in vitro [27] and Ad5-mediated gene transfer in vivo [28]. Together, this suggests that repeat administration of RGD-modified oncolytic adenovirus could be efficacious, in particular, to treat cancer in the brain, where we found that AdLucRGD-neutralizing activity remained very low. Hence, if RGD-modified oncolytic adenovirus is used, switching between capsid variants or serotypes, as previously suggested, to increase the efficacy of repeat administration [29,40] might not be necessary. Notably, we observed a discrepancy between total anti-Ad5 IgG titers and anti-Ad5 NAb titers. Hence, the presence of anti-adenovirus antibodies should not be misinterpreted as NAbs hampering oncolytic virotherapy.

In the course of our experiments, we noticed that the serum of many patients stimulated RGD-mediated adenovirus infection. This did not appear to be attributable to their disease or treatment, as also plasma from the majority of healthy donors had this property. Whilst human serum is known to contain factors, in particular, blood coagulation factors, that promote Ad5 infection of certain cell types such as hepatocytes [39], a serum factor specifically promoting RGD-mediated infection has, as far as we are aware, not been reported before. Therefore, we attempted to partially characterize this putative factor. We found that it was present in serum and plasma and that heat-inactivation augmented the effect if this was done on serum but not on plasma. Hence, the unknown factor that promotes RGD-mediated adenovirus infection is most probably not a blood coagulation factor, which is depleted from serum. Increased transduction was probably also not caused by inactivating neutralizing complement factors because these are also present in plasma. Instead, our observations showed that the yet-to-be-characterized factor is activated by mild heating to uncover its full potency in promoting the uptake of RGD-modified Ad5. This suggests that its activity is partially masked by a heat-labile inhibitory factor present in serum. The putative infection-promoting factor could be separated from neutralizing activity via size exclusion or by depleting the serum of IgG antibodies. Hence, the factor is not an antibody, excluding the possibility that the observed phenomenon could be explained by antibody-dependent enhancement of infection. The elution of the infection-promoting factor from the size exclusion column in many fractions did not reveal its MW. Perhaps it has a small MW and forms high-MW macromolecular complexes or is associated with extracellular vesicles. Full characterization of the blood component that promotes RGD-mediated adenovirus infection is beyond the scope of the present work. Identification of this molecule could, however, be worthwhile, as it might aid in predicting the success of oncolytic virotherapy with RGD-modified viruses.

## 4. Materials and Methods

### 4.1. Collection and Preparation of Human Serum, Plasma and Cerebrospinal Fluid Samples

Serum, plasma and CSF samples from patients with recurrent glioblastoma undergoing treatment with Ad5-∆24.RGD administered to the tumor and tumor-infiltrated surrounding brain by CED (ClinicalTrials.gov Identifier: NCT01582516) were collected, processed and stored as described [24]. Samples were collected before the start of the virus infusion and at 2 and 4 weeks after the start of infusion. In addition, the fluid fraction (i.e., rest material after isolation of peripheral blood mononuclear cells) of buffy coats obtained upon centrifugation of anticoagulated blood from consenting healthy donors (Sanquin Blood Supply Services, Amsterdam, The Netherlands) was used. Because in the PBMC isolation process, the buffy coats were diluted in PBS, these samples should be considered equivalent to 3-fold diluted human plasma.

For size exclusion chromatography, serum samples were heat-inactivated at 56 °C for 1 h and cleared from aggregates by centrifugation at 18,000× *g* for 5 min before being loaded onto a Sepharose 6B (Sigma-Aldrich, Zwijndrecht, The Netherlands) beaded agarose column equilibrated in PBS (0.5 mL serum on a 15 mL packed column). Elution was achieved by gravity flow with PBS, and 35 fractions of 0.5 mL were collected. Selected size-fractionated samples were depleted of IgGs using NAb Protein G Spin Columns, 0.2 mL (Thermo Scientific, Bleiswijk, The Netherlands). To this end, two consecutively eluted fractions from the Sepharose 6B column were pooled, and 150 µL was loaded onto the Protein G column equilibrated with BupH Phosphate Buffered Saline (0.1 M phosphate, 0.15 M sodium chloride, pH 7.2; Thermo Scientific). After incubation at room temperature with end-over-end mixing for 1 h, IgG-depleted samples were collected by centrifuging the column for 1 min at 5000× *g*.

### 4.2. Adenovirus Neutralization Assay

NAb titers were essentially determined as described before [41]. Unless specified otherwise, samples were heat-inactivated at 56 °C for 1 h before use in the assay. A549 human lung carcinoma cells (ATCC, Manassas, VA, USA) were seeded one day before infection at 1 × 10^4^ cells/well in 96-well plates and cultured overnight in Dulbecco’s modified Eagle’s medium (DMEM; Sigma-Aldrich) with 10% heat-inactivated fetal bovine serum (FBS; Gibco™, Grand Island, NY, USA) and antibiotics (100 U/mL penicillin and 100 µg/mL streptomycin; Sigma-Aldrich). Samples were diluted in DMEM in triplicate 2-fold dilution series, mixed 1:1 with AdLuc (originally designated AdCMV-Luc) [31] or AdLucRGD [32] adenoviral vector stocks prepared at 2-times final concentration and incubated for 1 h at RT. Next, the medium on the A549 cells was replaced by the virus/sample mixture, equaling 500 virus gc per cell. Control cultures received culture medium or 500 gc/cell virus diluted in culture medium. These controls served to define 0% and 100% infection efficiencies, respectively. After incubating the cells for 24 h at 37 °C and 5% CO_2_, cells were lysed, and luciferase activities were measured using the Luciferase Assay System (Promega, Madison, WI, USA). NAb titers were defined as the maximal sample dilution that reduced luciferase activity by at least 50%.

### 4.3. Statistical Analysis

All statistical analyses, using the appropriate tests specified in the description of the experiments, were done using GraphPad Prism version 10.2.0 (GraphPad Software, San Diego, CA, USA).

## 5. Conclusions

We found that treatment of human patients with RGD-modified oncolytic adenovirus induced mainly neutralizing antibody responses inhibiting adenovirus native tropism. Infection of cells with RGD-modified virus was not severely hampered by serum from these patients. Hence, the RGD modification of the viral capsid did not introduce an immune dominant epitope, and the RGD-mediated alternative cell entry route allowed to bypass pre-existing and induced neutralization by NAbs binding to wild-type capsid epitopes. The notable practical consequence of this finding is that RGD-modified oncolytic adenoviruses are probably more suitable for repeat administration treatments than adenoviruses with wild-type capsids. We also confirmed that adenovirus-neutralizing activity is very low in the brain. Moreover, while adenovirus administration into the brain induced neutralizing activity in the blood, this remained low in the brain. An unexpected side-observation of our work was the discovery of RGD-modified adenovirus infection-promoting activity in the blood of most humans. In most but not all cases, this activity was masked by the concomitant presence of neutralizing antibodies. It only became apparent upon removal of immunoglobulins by dilution, size fractionation or IgG depletion. The nature of the serum component that promotes cellular uptake of RGD-modified adenoviruses is currently unknown. Together, our work supports the utility of RGD-modified oncolytic adenoviruses for the treatment of cancer in humans.

## Figures and Tables

**Figure 1 ijms-26-00854-f001:**
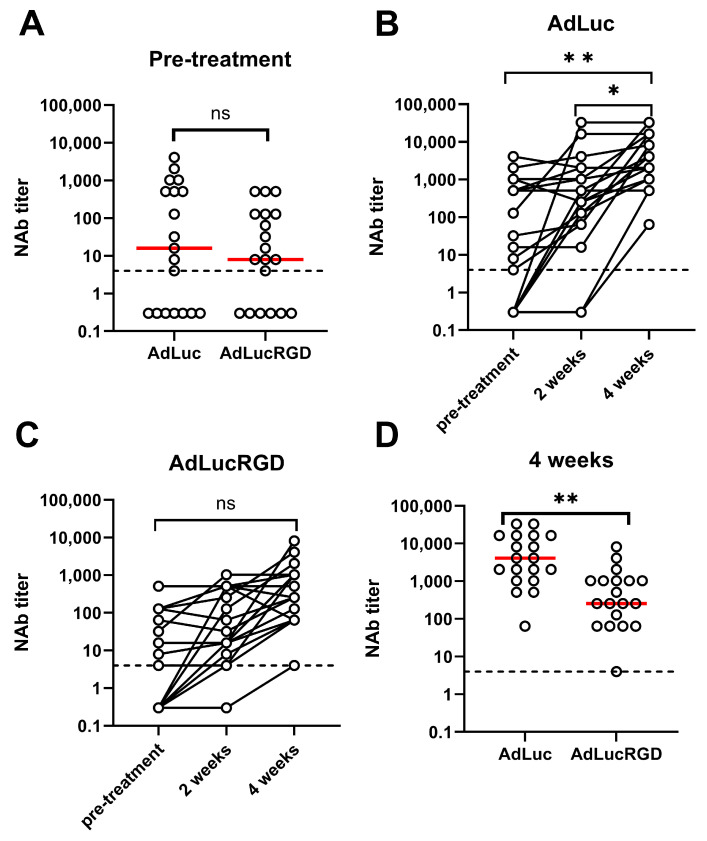
Adenovirus neutralizing antibody titers in the serum of patients with recurrent glioblastoma undergoing treatment by intracranial infusion of oncolytic adenovirus Ad5-∆24.RGD. Titers are derived from the serum dilution titration experiments shown in Appendix A. The assay detection limit (4) is shown in all panels (dashed line). Samples with undetectable NAb titer are shown at arbitrary positions below the detection limit. (**A**) Pre-treatment titers of antibodies that neutralize AdLuc and AdLucRGD. (**B**) Comparison of titers before and at 2 and 4 weeks after Ad5-∆24.RGD infusion, for antibodies that neutralize AdLuc. (**C**) Comparison of titers before and at 2 and 4 weeks after Ad5-∆24.RGD infusion, for antibodies that neutralize AdLucRGD. (**D**) Titers of antibodies that neutralize AdLuc and AdLucRGD, 4 weeks after Ad5-∆24.RGD infusion. Individual data points for all 19 patients are shown in all panels, with medians in panels A and D indicated with red lines. Statistical tests used are paired two-way *t*-test in (**A**,**D**) and One-Way ANOVA on matched data, with Tukey’s multiple comparisons test in (**B**,**C**). ns, not significant; *, *p* < 0.05; **, *p* < 0.01.

**Figure 2 ijms-26-00854-f002:**
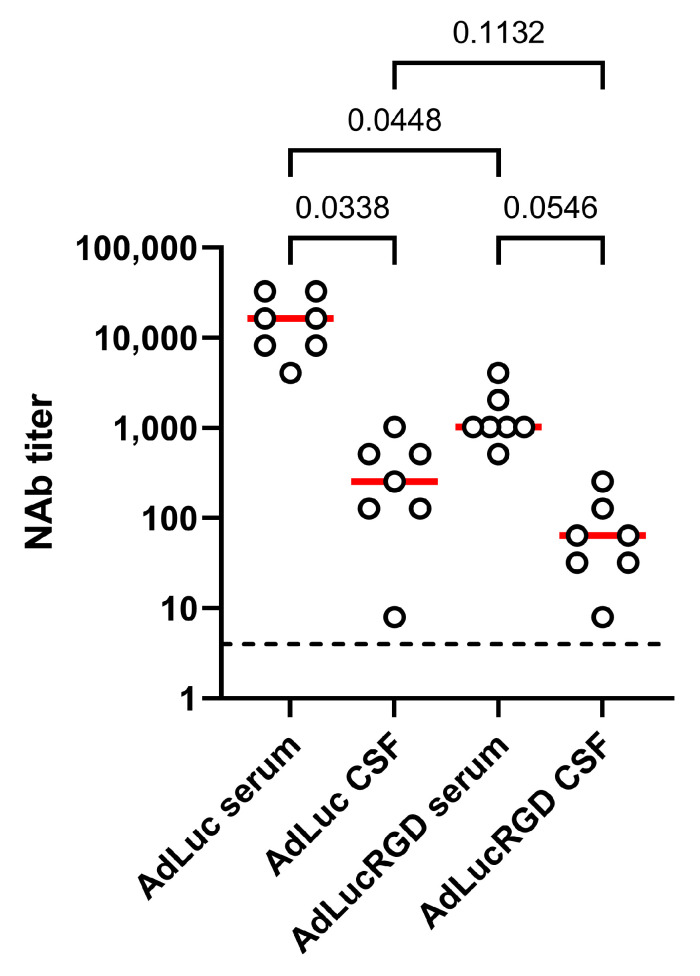
Reduced adenovirus NAb titers in cerebrospinal fluid. Titers of antibodies that neutralize AdLuc and AdLucRGD were determined in the CSF of eight glioblastoma patients who had been treated 4 weeks before with oncolytic adenovirus Ad5-∆24.RGD. Measured titers are compared with matching serum samples from the same patients collected on the same day. The graph shows the individual data (dots) with medians (red lines). The assay detection limit (4) is indicated by the dashed line. Statistical differences were tested using One-Way ANOVA on matched data, with Tukey’s multiple comparisons test. *p*-values for compared groups are shown.

**Figure 3 ijms-26-00854-f003:**
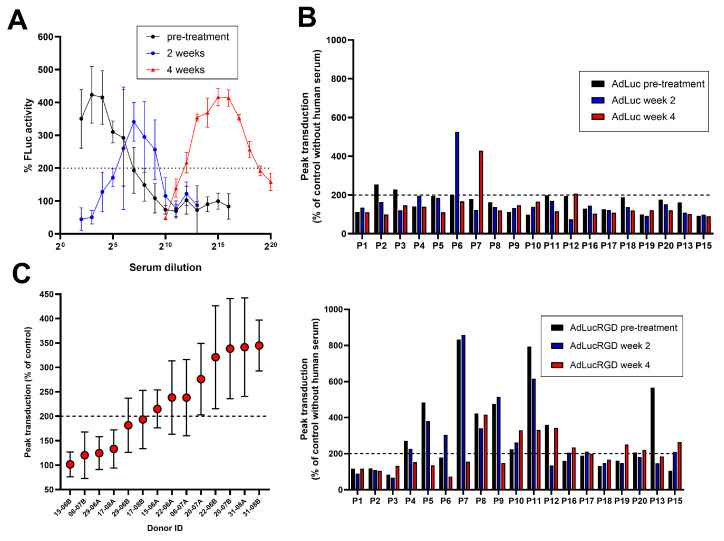
Identification of blood samples that promote adenovirus vector-mediated transduction. (**A**) Serum dilution titration experiment with AdLucRGD on samples from patient #8. (**B**) Peak transduction was reached with AdLuc (upper panel) or AdLucRGD (lower panel) for all patient serum samples relative to controls without human serum. (**C**) Peak transduction with AdLucRGD that is reached in the presence of plasma from 14 healthy donors relative to controls without human plasma (mean ± SD). In panel C, samples are ranked according to increasing transduction efficiency. In all panels, the reference line at 200% transduction that was used as an arbitrary threshold to score positive samples is shown (dashed line).

**Figure 4 ijms-26-00854-f004:**
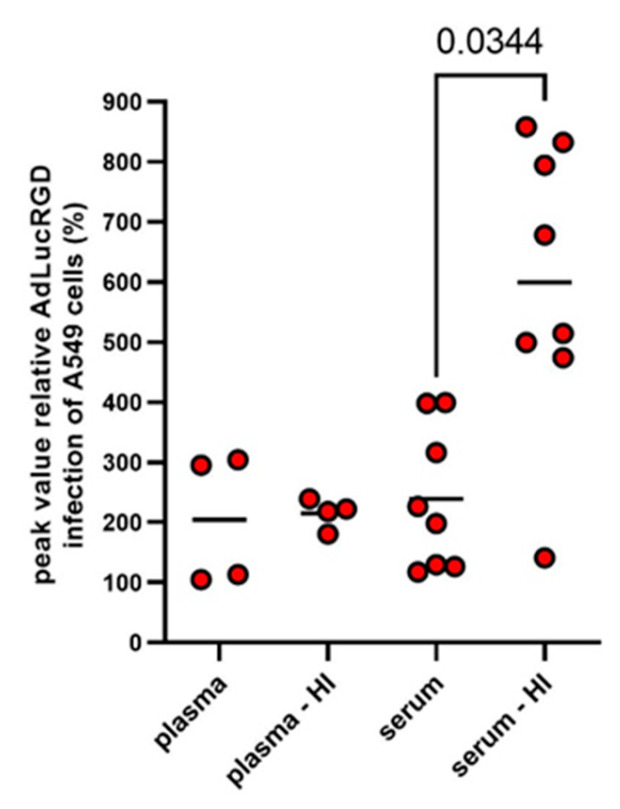
Comparison of RGD-modified adenovirus infection-promoting activities in human blood products. Blood samples from glioblastoma patients who were treated with oncolytic adenovirus were used to isolate serum or plasma. Part of these samples were heat-inactivated before use. The dilution series of these samples were tested for the transduction efficiency of AdLucRGD on A549 cells. Data shown are the peak transductions reached for all samples at any dilution, relative to controls without human serum (dots) and means per group (lines). Matched samples from eight patients were used. Plasma samples were available from only four of these patients. Therefore, the unbalanced datasets were compared by a Restricted Maximum Likelihood (REML) Mixed-effects model on matched data. Only the significant difference identified by Tukey’s multiple comparisons test is shown.

**Figure 5 ijms-26-00854-f005:**
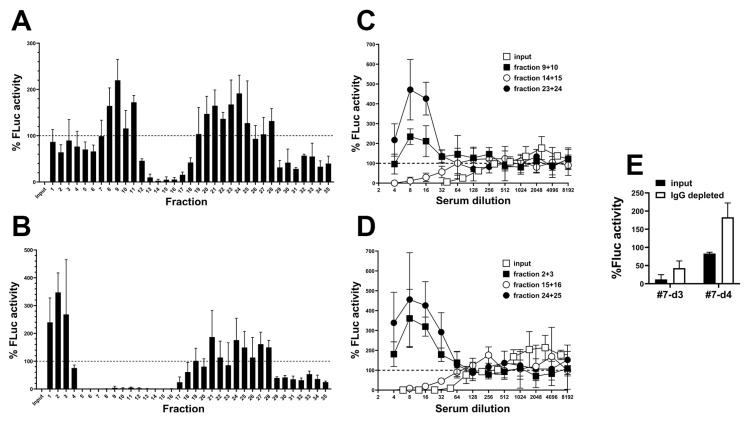
Separation of RGD-modified adenovirus neutralizing and infection-promoting activities in human serum by size exclusion and IgG depletion. Sera from patient #7 collected 3 (**A**,**C**,**E**) and 4 (**B**,**D**,**E**) days after oncolytic adenovirus infusion were separated over a Sepharose 6B column. (**A**,**B**) Individual fractions were tested for their influence on A549 cell transduction by AdLucRGD. Collected fractions were tested at 5-fold dilution, in comparison to AdLucRGD without serum (100%) and AdLucRGD with unfractionated 5-fold diluted serum (input). Data are the means + SD from a single experiment performed in triplicate. (**C**,**D**) Full AdLucRGD transduction neutralization analysis of selected pooled low, medium and high-MW fractions in comparison to unfractionated serum (input). Fractions were selected on the basis of the elution profiles observed in panels A and B. Data are the means ± SD from a single experiment performed in triplicate. (**E**) Transduction neutralization analysis of pooled medium MW fractions (i.e., day-3 fractions 14 and 15; day-4 fractions 15 and 16) before and after IgG depletion, tested at 8-fold dilution (i.e., peak transduction observed in panels (**C**,**D**)). Data are the means + SD of two experiments performed in quintuplicate.

## Data Availability

All data analyzed in this study are included in this published article and its Appendix A.

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
