# Peer review of "Adenovirus-Neutralizing and Infection-Promoting Activities Measured in Serum of Human Brain Cancer Patients Treated with Oncolytic Adenovirus Ad5-∆24.RGD"

_ijms, 2025, doi:10.3390/ijms26020854_

Round 1
Reviewer 1 Report
Comments and Suggestions for Authors
In their study, van der Meulen-Muileman and colleagues evaluated the humoral immune response to the oncolytic Ad5-∆24 viral platform using clinical samples from glioblastoma patients undergoing infusion-based treatment. Specifically, the authors focused on the impact of both pre-existing and treatment-induced neutralizing antibodies (NAbs) on the efficacy of the virus. The results demonstrated that while NAbs predominantly neutralized the virus's native tropism, the RGD modification in Ad5-∆24 allowed the virus to bypass this neutralization mechanism. This modification significantly enhanced the virus’s ability to infect target cells despite the presence of NAbs. Interestingly, in cerebrospinal fluid, the neutralizing activity against the RGD-mediated infection remained minimal, suggesting that the alternative cell entry route via integrins conferred a degree of resistance to immune-mediated clearance.
Furthermore, the study identified that human serum actively promotes the uptake of the RGD-modified adenovirus into human cells, pointing to an as-yet unidentified infection-stimulating factor that may enhance viral delivery.
Overall, the findings support the potential of RGD-modified oncolytic adenoviruses , particularly for patients requiring repeat administrations, since these modified viruses are less susceptible to neutralization by antibodies.
This research highlights the strategic advantage of incorporating RGD modifications into oncolytic adenoviruses to overcome immune barriers and improve treatment efficacy. The manuscript is well-organized, with clear experimental design and good presentation of the results. The methodology is correct, and the findings offer valuable insights into how oncolytic adenoviruses can be optimized for clinical application.
Minor Comments/Revision: The authors should provide evidence that the observed enhancement of infection is not specific to the A549 cell line. Performing the assay in a different cell line would help to further support and validate their findings, ensuring that the observed effect is not cell-line dependent.
Author Response
See uploaded pdf.

Reviewer 2 Report
Comments and Suggestions for Authors
Dear authors, your manuscript addresses an important aspect of oncolytic virotherapy, that is, evasion of immune responses. While you have done a lot of work trying to demonstrate that Ad5-Δ24.RGD induces less neutralizing antibodies than the native Ad vector, the results in respect of mechanism need further speculation. One should address the possible impact of inclusion of RGD to virus vector and to the lack of neutralizing epitopes, are any known? The latter part of the study, although interesting, is mainly negative results, and lacks the concept of antibody-mediated enhancement of infection (ADE). I wonder if you could compare the pre-existing total Ad values to enhancement and at least to exclude their role in enhancement. The paper needs more discussion about the enhancement since it is not unknown factor in virus biology, though the mechanism is unclear.
Specific comments:
Figure 1 is difficult to read. The subpanels B and C need to be separated from each other to clarify the results for Ad-Luc and AdLucRGD in B. Please, re-modify the image.
The rationale (Fig. 2) behind analyzed of CSF antibodies needs further clarification besides citing one´s own results. CNS normally has low antibody amounts and thus it would serve the audience to explain early enough why such analyses were performed.
One should discuss of tropism, Ad antibody epitopes in respect of vector structure. Refs 24 and 27 should be explained in the context of this paper.
Author Response
see uploaded pdf.

Reviewer 3 Report
Comments and Suggestions for Authors
· Summary
This manuscript discusses oncolytic adenoviruses, specifically Ad5-∆24.RGD, which have modified fiber proteins and show potential for cancer treatment by overcoming pre-existing neutralizing antibodies (NAbs) in glioblastoma patients. The findings indicate that these modified viruses are less impacted by NAbs, facilitating more effective cell entry and infection. Furthermore, human serum appears to enhance the uptake of RGD-modified adenoviruses, suggesting their viability for repeated use in cancer therapies. The manuscript is well-organized and includes data from significant patient samples; however, I recommend that the authors carefully address the following points.
· Major issues
1. In Figure 3A, it is observed that the virus infection rate decreases with higher dilutions after an initial increase. Typically, one would expect that the efficacy of infection would continue to rise with increasing dilutions. Please provide a detailed explanation for this phenomenon.
2. The authors reference an infection-stimulating factor but do not provide any explanation regarding its nature or function. It is essential for the authors to elaborate on this factor or discuss potential factors that may contribute to this effect, including any predictions based on existing literature.
3. While the manuscript contains sentences that are grammatically correct, the language used is somewhat difficult to read and does not conform to the standard conventions of scientific writing, particularly in the field of biology. A thorough review and revision of the manuscript for clarity and adherence to scientific language norms is recommended.
· Minor issues
1. In the manuscript, please change "preexisting" to "pre-existing."
2. On page 1, the phrase "The for now unidentified..." lacks clarity. It would be beneficial to rephrase this for better understanding.
3. On page 3, in the sentence "To investigate if the antibodies...," it is important to clarify that the object of neutralization is adenovirus, not adenovirus infection. Therefore, "neutralizing adenovirus infection" should be revised to either "neutralizing adenovirus" or "decreasing adenovirus infection." Please ensure that the entire manuscript is reviewed for consistency.
4. On page 3, please modify "coxsackievirus and adenovirus receptor" to "coxsackievirus and adenovirus receptor (CAR)" for clarity and to introduce the abbreviation properly.
Comments on the Quality of English Language1. While the manuscript contains sentences that are grammatically correct, the language used is somewhat difficult to read and does not conform to the standard conventions of scientific writing, particularly in the field of biology. A thorough review and revision of the manuscript for clarity and adherence to scientific language norms is recommended.
Author Response
See uploaded pdf.

Round 2
Reviewer 2 Report
Comments and Suggestions for Authors
I thank the authors for prompt responses to this reviewer comments.
Reviewer 3 Report
Comments and Suggestions for Authors
Dear Authors,
I appreciate your efforts and comments in the response and the revised manuscript.
Best,